# Effects of Tomato Paste By-Product Extract on Growth Performance and Blood Parameters in Common Carp (*Cyprinus carpio*)

**DOI:** 10.3390/ani12233387

**Published:** 2022-12-01

**Authors:** Osman Sabri Kesbiç, Ümit Acar, Mohamed Shaban Hassaan, Sevdan Yılmaz, Maria Cristina Guerrera, Francesco Fazio

**Affiliations:** 1Department of Animal Nutrition and Nutritional Diseases, Veterinary Faculty, Kastamonu University, Kastamonu 37100, Turkey; 2Department of Forestry, Bayramiç Vocational School, Çanakkale Onsekiz Mart University, Çanakkale 17100, Turkey; 3Department of Animal Production, Fish Research Laboratory, Faculty of Agriculture at Moshtohor, Benha University, Benha 13736, Egypt; 4Department of Aquaculture, Marine Science and Technology Faculty, Çanakkale Onsekiz Mart University, Çanakkale 17100, Turkey; 5Zebrafish Neuromorphology Lab, Department of Veterinary Sciences, University of Messina, 98168 Messina, Italy; 6Department of Veterinary Sciences, Messina University, 98166 Messina, Italy

**Keywords:** feed additive, serum biochemistry, extract, lycopene, growth promoter

## Abstract

**Simple Summary:**

The purpose of this research is to evaluate the effects of a tomato paste by-product extract as a dietary supplement for common carp (*Cyprinus carpio*). At the end of a feeding trial, fish fed with experimental diets showed enhanced growth performance that is key for aquaculture, and a reduced feed conversion ratio, another critical parameter. Hematology parameters such as erythrocyte count (RBC), hemoglobin content (Hb), and hematocrit (Hct) increased in fish feed with the enriched diet. Additionally, blood glucose, cholesterol, and triglyceride levels in common carp fed diets supplemented with tomato paste by-product extract reduced while total protein, albumin, and globulin levels increased. In conclusion, tomato paste by-product extract has the potential to be utilized as a feed additive in aquaculture feeds to enhance health status monitored by blood parameters and stimulating the growth parameters of common carp.

**Abstract:**

This research examined different growth performances and blood parameters of common carp (*Cyprinus carpio*) consuming diets supplemented with tomato paste by-product extract (TPE). Five diets with different concentrations of TPEs, 0% (TPE0) 0.5% (TPE0.5), 1% (TPE1), 2% (TPE2), and 5% (TPE5) were supplied to 300 common carp (8.38 ± 0.18 g) (60 fish per group) each day during the 60-day feeding trial. The relative and specific growth rates of fish-fed supplemental diets increased significantly, while the feed conversion ratio significantly decreased (*p* < 0.05). TPE considerably increased the erythrocyte count (RBC), hemoglobin content (Hb), and hematocrit (Hct) (*p* < 0.05), according to hematological analyses of blood samples collected after the feeding experiment. The blood biochemical findings indicate that using 1% or more extracts considerably reduced the serum glucose, cholesterol, and triglyceride ratios while significantly increasing the total protein, albumin, and globulin ratios in common carp (*p* < 0.05). Based on the findings of the study, it was concluded that the 2% extract generated from tomato paste by-products in common carp diets could be utilized as a growth-promoting product without any negative effects on blood parameters linked to feeding in carp.

## 1. Introduction

Recently, aquaculture has become a developing food production sector with an approximate 8% average yearly growth rate that has been achieving animal protein requirements by compensating for the deficiency in meat production [1]. Currently, to achieve large amounts and sustainable production of fish, an intensive aquaculture system must be applied on big farms [2,3,4]. Furthermore, intensive aquaculture is simply the use of high densities of fish stocking to produce high productivity while using minimal water resources [5,6]. However, intensive aquaculture systems may be faced with big challenges caused by the quality of the water environment and outbreaks of diseases, which result in significant financial losses. For this reason, experts are interested in testing new active biological substances that can be potentially used in aquatic feed to achieve aquaculture sustainability [7,8,9]. Large amounts of by-products comprising various reusable and high-value components with economic potential are generated during the industrial processing of fruits and vegetables. These by-products are high in bioactive chemicals such as phenolic compounds, flavonoids, carotenoids, and anthocyanins, which can have anticancer, antiviral, antitumor, antibacterial, and antioxidant effects [10,11,12,13]. Much of the research on the utilization of plants and plant extracts in aquaculture practices has been performed due to the benefits supplied by the bioactive components contained within plants [14,15,16]. Many fruits and vegetables, including tomatoes and their processed derivatives, watermelon (*Citrullus lanatus*), guava (*Psidium guajava*), carrots (*Daucus sativus*), and pink grapefruit (*Citrus paradisi*), contain lycopene, a red-colored pigment. Lycopene is one of the most effective antioxidants due to it containing large amounts of conjugated dienes. Its capacity to quench singlet oxygen is twice as great as that of carotene and ten times higher than that of tocopherol [17,18]. Along with these qualities, several studies have demonstrated the crucial role of lycopene in enhancing gene regulation, immunological functions, hormonal and metabolic pathways, and protecting against medical problems including cancer, cardiovascular, and degenerative diseases [19,20]. Although lycopene is the primary focus of study on tomatoes, tomato products and wastes, tomatoes also contain many other bioactive components such as volatile compounds and flavonoids [21]. All these bioactive components, especially lycopene, have shown that tomato products and tomato extracts have many benefits, especially high antioxidant effects [22,23].

Although studies on the use of tomato-derived products and extracts in the nutrition of livestock for different purposes, limited knowledge has been reported the effects of wastes of tomato-using industries on fish. Therefore, this study focuses on investigating the potential effects of tomato paste by-product extract on growth, feed utilization, hematology, and serum biochemical indices of common carp (*Cyprinus carpio*).

## 2. Materials and Methods

A total of 300 juvenile common carp (8.38 ± 0.18 g) were randomly allocated into aquariums with 20 fish per aquarium for the feeding experiment. The fish were fed the experimental diet by hand for 60 days to apparent satiation and a total of 15 aquariums (100 L) were used in the experimental setup to provide five dietary treatments in triplicates. During the feeding experimentation period, the fish were held at ambient temperatures of 24 ± 1.5 °C, dissolved oxygen of 6.03 ± 0.51 mg L^−1^, pH of 7.42 ± 0.68, nitrate 1.5 ± 0.6 mg L^−1^, nitrite 0.03 ± 0.00 mg L^−1^ and ammonia 0.04 ± 0.00 mg L^−1^. This experiment was authorized by Çanakkale Onsekiz Mart University’s local ethical committee who granted approval for this study (ref. 2017/04-11). Financial support for this study was provided by the Scientific Research Project Fund of Kastamonu University (Project No: KÜ-BAP01/2019-2).

### 2.1. Preparation of Tomato Paste By-Product and Extraction

The tomato pulps used in this study were obtained from a commercial tomato paste factory in Turkey. The pulps were separated from the seeds and dried in a fan assisted oven (Nüve FN 120, Ankara, Turkey) under 70 °C then transported to the laboratory. Afterward, the pulps were powdered with a grinder and kept in an ultrasonic bath for 30 minutes at 50 °C with a solvent mixture [hexane, acetone, ethanol (2:1:1)] containing 0.05% butyl hydroxyl toluene (BHT) at a ratio of 1:30. After this step, the mixture was filtered with Watman 90 mm circular filter paper. Organic solvents were removed from the filtrate by rotary evaporator and the tomato paste by-product extract (TPE) was achieved [24]. The extract was stored in amber bottles with nitrogen gas and stored at −20 °C until it was analyzed and used in the experimental diets.

### 2.2. Determination of Lycopene and Volatile Component Content of the Tomato Paste By-Product Extract

The stored extract was dissolved with a mixture of n-hexane, acetone, and ethanol (2:1:1 (v:v:v)) and then the solution was measured with a spectrophotometer (WTW Photo Lab 7600, Weilheim, Germany) at 503 nm wavelength by 3.5 mL standard quartz spectrophotometer cuvette. The amount of extract lycopene was calculated as mg/100 g according Davis et al. [25] and the volatile component profile of the extract was measured qualitatively by GC/MS. For the measurement, 0.1 g of the extract was deluded in 10 mL of n-hexane and then injected into the GC/MS (Shimadzu GCMS –QP 2010 ULTRA, Kyoto, Japan). During the analysis, the instrument condition was adjusted according to Kesbiç and Yiğit [26]. The peaks obtained after the analysis were compared with the W9N11 library and those with a similarity rate of more than 99% were considered significant.

The results showed that 97.32% of the extract's volatile component content could be identified. Cumene (38.78%), pyranton A (36.63%), and mesityl oxide (14.15%) accounted for the majority of this composition (Table 1). In addition, the lycopene content of the extract was determined as 5.67 mg 100 g^−1^.

### 2.3. Preparation of Experimental Diets

Dry materials were initially put into the laboratory-type mixer and mixed until homogenate was achieved in the production of experimental diets. TPE was added to the fish oils to be used in the experimental ration at the rates of 0%, 0.5%, 1%, 2%, and 5% of the total ration and these supplemented fish oils were used in the production of the experimental diets. The process was carried out to ensure a homogeneous distribution of TPE in the experimental diets. After the mixture was moistened with 10% de-ionized water, it was formed to a size of 1.6 mm in a pelletizing machine (Household AKK-125, Henan, China), dried at 40 °C and stored at −20 °C until the feeding trial. Analysis of the nutritional contents of the experimental diets such as moisture (AOAC, 934.01), protein (N × 6.25; AOAC, 955.04), fat (AOAC, 954.02), and ash (AOAC, 942.05) were carried out as percentages based on the standard methods. The formulation and nutritional content of experimental diets were represented in Table 2.

### 2.4. Feeding Experiment and Determination of Growth Performance

At the beginning of the study, each fish was weighted (initial weight) and then placed in the tanks (100 L) and hand-fed three times a day (at 8:30, 12:00, and 17:00) for 60 days till apparent satiation. At the end of the experiment, the fish in the groups were weighed again (final weight), and growth and feed utilization performances were calculated using the weights of the fish and diets at the beginning and end of the experiment by using the following formulas.
(1)$$\mathrm{Feed}\;\mathrm{Conversion}\;\mathrm{Rate}\;(\mathrm{FCR})=\frac{\mathrm{Feed}\;\mathrm{Consumed}\;\left(g\right)}{\mathrm{Wet}\;\mathrm{Weight}\;\mathrm{Gain}\;\left(g\right)}$$
(2)$$\mathrm{Relative}\;\mathrm{Growth}\;\mathrm{Rate}\;(\mathrm{RGR})=\frac{\mathrm{Wet}\;\mathrm{Weight}\;\mathrm{Gain}\;\left(g\right)\;}{\mathrm{Initial}\;\mathrm{Weight}\;\left(g\right)}\times 100\mathrm{ss}$$
(3)$$\mathrm{Specific}\;\mathrm{Growth}\;\mathrm{Rate}\;(\mathrm{SGR})=\frac{\left[\mathrm{Ln}\left(\mathrm{Final}\;\mathrm{Weight}\right)-\mathrm{Ln}\left(\mathrm{Initial}\;\mathrm{Weight}\right)\right]}{\mathrm{Experimental}\;\mathrm{Days}}\times 100$$

### 2.5. Collection and Analysis of Blood Samples from Fish

Following the feeding study, blood samples from 12 fish per group, 4 from each aquarium were used for hematological and biochemical analyses. The fish were anesthetized with a clove oil and ethanol mixture (1:5 (v:v)) in an immersion bath (25 mg/L), a natural product and a common anesthetic used in fisheries [27]. Blood was collected from the caudal vein with a syringe needle as soon as possible after thoroughly cleaning the sampling place with alcohol to prevent mucous from getting into the blood. The blood samples totally 12 fish from each experimental group were equally divided in lavender-topped blood collection tubes for hematological analysis, and gold-topped tubes for biochemical analysis. Hematological analyses were performed immediately after blood collection: gold-topped tubes were centrifuged at 5000 g for 10 min under +4 °C to separate the serum useful for the measurement of biochemical parameters. The serum was stored at −86 °C until analysis.

#### 2.5.1. Hematological Analyses

Hematological parameters such as erythrocyte count (RBC), hematocrit (Hct), and hemoglobin (Hb) concentration were determined using the methods described by Blaxhall and Daisley [28]. The RBC count was counted using Dacie’s diluting solution and a Thoma hemocytometer. The Hct value was determined using a capillary Hct tube and hematocrit centrifuge (+4 °C and 5000 g). The Hb concentration was measured by using the cyanmethemoglobin technique by Drabkin’s reagent (D5941; Sigma-Aldrich, Missouri, USA) and spectrophotometry (540 nm).

#### 2.5.2. Serum Biochemical Analyses

The spectrophotometric method by microplate reader (Epoch 2, BioTek, VT, USA) and with commercial kits (Bioanalytic Diagnostic Industry, İstanbul, Turkey) previously applied in fish was used to analyze the parameters of GLU, albumin (ALB), globulin (GLO), total protein (TPORT), triglyceride (TRIG), cholesterol (CHOL), lactate dehydrogenase (LDH), alkaline phosphatase (ALP), and glutamic-pyruvic transaminase (GPT) [29].

### 2.6. Statistical Analysis

After evaluating the data normality and variance homogeneity, one-way ANOVA was utilized to compare the groups. If the ANOVA was significant, the post-hoc Tukey's test was used to compare the treatment means. The data are presented in the form of means and standard deviations. SPSS 16.0 statistical software was used to perform all statistical analyses and summary statistics (IBM, Armonk, NY, USA).

## 3. Results

### 3.1. Growth Performance and Feed Utilization Parameters

At the end of the 60-day feeding study, all groups fed with TPE-supplemented diets had significantly improved growth and feed conversion performance compared to the control group (TPE0) (*p* < 0.05). Although the highest growth performance and the lowest FCR were found in fish fed with TPE2 group diet, no significant differences were observed between the groups fed with TPE-supplemented feed (*p* > 0.05) (Table 3).

The relationship between the relative growth rate (RGR) and the TPE concentration used in the diets, showed that the best possible growth performance would be observed in common carp fed with 3.11% TPE-supplemented feeds (Figure 1).

### 3.2. Hematological Parameters

TPE supplementation to common carp diets significantly increased all hematological parameters such as RBCs, Hb, and Hct in fish fed with all experimental groups (*p* < 0.05) (Table 4.).

### 3.3. Serum Biochemical Parameters

Measurements of serum glucose, total protein, albumin, globulin and lipid parameters such as cholesterol, and triglycerides were significantly affected by adding TPE compared with the control group (*p* < 0.05). The blood-liver enzyme markers AST, ALT, ALP, and LDH tended to decrease, although the changes were not statistically significant (*p* > 0.05) (Table 5.). Serum glucose levels decreased with TPE supplementation up to a 2% rate, but there was no significant difference in serum glucose levels between fish fed 2% (TPE2) and 5% (TPE5) supplemented feeds. The serum protein values of common carp consuming different ratios of TPE through feed showed an increasing trend depending on the TPE ratio in the feed. Serum total protein content of common carp consuming 1, 2, and 5% TPE significantly increased (*p* < 0.05) compared to the control group, while no significant difference was observed in those consuming 0.5% TPE compared to the control group (TPE0) (*p* > 0.05). The supplementation level of TPE led to a significant increase on serum albumin value (*p* < 0.05). While TPE supplementations up to 1% were increased serum albumin value, 2 and 5% were not rise albumin further. There was no significant difference in serum globulin value between the TPE supplemented groups and the control group. However, the globulin value of the fish fed TPE0.5 group feed was significantly lower than that of the fish fed TPE1 and TPE2 group diets (*p* < 0.05). Serum lipid values such as cholesterol and triglycerides started to decrease significantly with the supplementation level of TPE. Although 0.5% supplementation did not significantly affect the serum triglyceride values compared to the control (*p* > 0.05), 1% and above supplementation significantly decreased this parameter (*p* < 0.05). A similar trend was observed in cholesterol levels with a significant decrease in this value starting with 0.5% supplementation (*p* < 0.05).

## 4. Discussion

This current study showed that the supplementation of tomato paste by-product extract (TPE) to common carp diets had a significant effect on growth and feed conversion performance with calculated RGR, SGR, and FCR values. Although there was no statistical difference in the growth and feed conversion parameters of the fish fed with different ratios of TPE-supplemented diets, the highest relative growth rate and the lowest feed conversion ratio were observed in carp fed with 2% supplemented diets. However, based on the relationship between relative growth rate and diet TPE concentration, it was determined that 3.11% supplementation provided the best growth performance (Figure 1). Previous research has shown that carotenoids, particularly lycopene from tomatoes, are beneficial in reducing stress caused by environmental conditions, chemotherapeutic agents, and/or physical effects on fish and preventing related performance falls. [30,31]. An enhancement in performance was also found in common carp fed a TPE-supplemented diet containing lycopene in the current investigation without stress factor. Previous studies on different fish species showed that carotenoids increase the secretion of some digestive enzymes, especially lipase and trypsin [32] and increase the abundance of probiotic bacteria especially *Bacillus* of the intestine [33]. Similarly, the dietary lycopene increased feed intake and improved the growth performance of rainbow trout (*Oncorhynchus mykiss*) [34]. In contrast to our findings, no significant variations in growth performance of goldfish (*Carassius auratus*) and platy fish (*Xiphophorus maculatus*) supplied diets enriched with tomato extract and lycopene were identified [35]. In addition, lycopene supplementation did not significantly change the growth performance and pigmentation of goldfish (*Carassius auratus*) [36]. These differences might be attributed to several factors such as the different fish species, feeding periods, experimental conditions, the physiological status of fish, and the specific formula of the diet or others. However, it is suggested that the differences are related to the products utilized in the current study. The majority of the study with tomato by-product extracts and/or tomato waste extract is centered on lycopene, known to be effective and whose usefulness has been extensively examined. In contrast to this study, the extract produced for this investigation contained volatile components. As a result, the benefits provided by the extract are assumed to be attributable to the synergistic impact of more than one component.

Hematological indices, red blood cell (RBC) count, hemoglobin (Hb) value, and hematocrit (Hct) are considered valuable indices to estimate fish health and the efficiency of feed [37,38]. In the current study, RBCs, Hb values, and Hct were significantly improved in fish fed with diets supplemented with different levels of TPE. Similarly, hematological parameters have also been developed in albino rats orally fed with 2 ml of tomato juice [39]. Moreover, a study reviewing the effect of lycopene on stress factors in fish [31] reported an improvement in hematological parameters in different species of fish that were fed diets supplemented with lycopene under different chronic stress conditions. Increase in haematological parameters such as RBC, Hct and Hbmight be associated with the antianemic effects of TPE [39] and it is caused by the main metabolite lycopene [40]. Further studies are needed to better investigate the erythropoiesis effect of TPE or lycopene on fish.

Serum biochemical parameters are functional tools to understand health and the physiological status of fish. For example, the primary source of energy for fish is glucose and the glucose is a biomarker of physiological stress. In addition to this the activity levels of serum aspartate aminotransferase (AST) and alanine aminotransferase (ALT) represent critical indicators in the diagnosis of digestive function and liver integrity [41]. The current study found that dietary TPE supplementation at various dosages might regulate blood glucose and AST levels in common carp. These findings are consistent with those reported by Mekkawy et al. [42], that blood glucose, AST, and ALT levels were considerably decreased in cadmium-exposed tilapia (*Oreochromis niloticus*) fed an experimental diet supplemented with containing 9 mg lycopene/kg diet. The recent publication shed light on the positive uses of tomato paste, tomato juice, and/or lycopene products in aquaculture for exceptional hepatoprotection by reducing blood AST and ALT levels [31]. The use of lycopene containing products extracted from tomatoes and their wastes in fish has been reported to have a liver protective effect even in cases of exposure to highly toxic heavy metals such as cadmium and may be effective in the regulation of stress due to various sources, as can be understood from selected prior studies [35,42]. That the GLU value, which is a stress indicator, decreased in fish fed with feeds containing TPE compared to the control group in the current study. In addition, liver enzymes of experimental groups were not significantly different from those of the control group. This finding shows that in the absence of liver damage due to stress and/or different toxic products, TPE can be safely used for its other beneficial outcomes without any negative effect on the liver.

Monitoring the changes in cholesterol and triglycerides is useful for showing the lipid metabolism as well as the health status of fish [43]. Our results are in agreement with the findings of [44], in which dietary lycopene decreased serum total cholesterol levels of fish exposed to artificial oxidative stress generated by ZnO Nps. Therefore, our investigation suggests that the anti-oxidative property of lycopene and other antioxidants in TPE such as cumene, pyranton A, and mesityl oxide (Table 2) may alleviate hyperlipidemia through the rise of endogenous cholesterol and triglyceride clearance mechanisms, which consequently relieve stress in fish [45]. The anti-hyperlipidemic potential of various types of tomato extracts and lycopene have been studied in different fish species with promising findings such as the ability to suppress oxidative stress caused by hyperlipidemia of outcomes [31]. Lycopene is a predominant carotenoid that gives tomatoes their red color [46], and is a strong antioxidant that potently reduces oxidative damage of lipids, proteins, and DNA [17,20]. Serum total protein, albumin, and globulin play important roles in the immune response in fish [47]. 

The results of this study show that increasing total protein, albumin, and globulin in fish fed with varying levels of TPE, reflected the strengthened immunity of fish. Unfortunately, studies are rarely found concerning the effect of TPE supplementation on serum proteins in aquatic animals [31]. Hassan et al. [48] noted that plasma protein and globulin levels were high in rabbits fed TPE-supplemented diets. Since it is well known that serum or plasma proteins are synthesized in the liver [49], the findings of the current study pointed that the antioxidant potential of TPE may have liver-protective and protein synthesis-promoting effects. The serum protein findings of the current study support the results of previous studies using tomato peel extract in rabbits [48]. 

## 5. Conclusions

This study indicated that TPE significantly improves growth performance, nutrient utilization, and hematological and serum biochemistry parameters. Based on the results of this study, analysis of physiological functions are recommended for further explanation of the mechanisms of lycopene and other antioxidants in tomato paste by-product extract.

## Figures and Tables

**Figure 1 animals-12-03387-f001:**
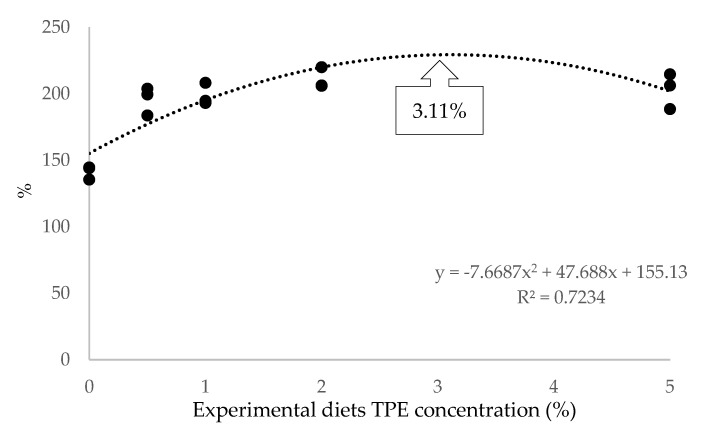
The relationships between dietary tomato paste by-product extract (TPE) concentrations and relative growth rate (RGR) (%) of *Cyprinus carpio.*

**Table 1 animals-12-03387-t001:** Volatile compounds of tomato paste by-products extract.

	Content	Retention Time (min.)	Concentration (%)
1	Mesitil oxide	4.681	14.15
2	Pyrrolidine	4.796	1.64
3	Pyranton A	5.988	36.63
4	Ethylbenzene	6.526	0.60
5	Styrene	7.567	0.35
6	Cumene	30.404	38.78
7	Fitone (6,10,14-Trimethylpentadecan-2-one)	39.898	0.29
8	1.2-Dipalmitin	50.763	0.85
9	Cholesterilene	54.129	1.93
10	(Z)6,(Z)9-Pentadecadien-1-ol	54.562	1.14
11	Monoelaidin	54.671	0.96
Total			97.32

**Table 2 animals-12-03387-t002:** Formulation and nutrient content of experimental diets supplemented with different ratios of tomato paste by-product extract (TPE).

Contents (g)	TPE0	TPE0.5	TPE1	TPE2	TPE5
Fish meal	190	190	190	190	190
Soybean meal	300	300	300	300	300
Corn starch	190	189.5	189	188	185
Wheat flour	200	200	200	200	200
Fish oil	80	80	80	80	80
Vit. Min. mix.^1^	40	40	40	40	40
TPE	-	0.5	1	2	5
Total	1000	1000	1000	1000	1000
Crude protein	301.67	301.60	301.66	301.71	301.16
Crude fat	104.06	104.10	104.09	104.07	104.09
Crude ash	46.30	46.30	46.30	46.30	46.30
Crude cellulose	31.35	31.34	31.33	31.32	31.27
NFE^2^	507.94	507.44	506.94	505.95	502.96

^1^ Vitamin Mineral Mixture: Fe. 75.3 mg kg^−1^ feed; Cu. 12.2 mg kg^−1^ feed; Mn. 206 mg kg^−1^ feed; Zn. 85 mg kg^−1^ feed; I. 3 mg kg^−1^ feed; Se. 0.350 mg kg^−1^ feed; Co. 1 mg kg^−1^ feed, Vitamin A. 18000 IU kg^−1^ feed; Vitamin D_3_. 2500 IU kg^−1^ feed; Vitamin E. 250 mg kg^−1^ feed Vitamin K_3_. 12 mg kg^−1^ feed; Vitamin B_1_. 25 mg kg^−1^ feed; Vitamin B_2_. 50mg kg^−1^ feed; Vitamin B_3_. 270 mg kg^−1^ feed; Vitamin B_6_. 20 mg kg^−1^ feed; Vitamin B_12_. 0.06 mg kg^−1^ feed; Vitamin C. 200 mg kg^−1^ feed; Folic acid. 10 mg kg^−1^ feed; Calcium d–pantothenate. 50 mg kg^−1^ feed; Biotin. 1 mg kg^−1^ feed; Inositol. 120 mg kg^−1^ feed; Choline chloride. 2000 mg kg^−1^ feed; ^2^ NFE = nitrogen free extract = 100 − (crude protein + crude fat + crude Ash).

**Table 3 animals-12-03387-t003:** Growth and feed utilization performance of common carp fed with experimental diets supplemented with different ratios of tomato paste by-product extract (TPE).

	TPE0	TPE0.5	TPE1	TPE2	TEP5	*p*-Value
Initial weight (g)	8.35 ± 0.18	8.35 ± 0.23	8.52 ± 0.15	8.35 ± 0.23	8.35 ± 0.22	0.804
Final weight (g)	20.15 ± 0.80 ^b^	24.66 ± 0.45 ^a^	25.43 ± 1.07 ^a^	25.93 ± 0.86 ^a^	25.28 ± 0.59 ^a^	0.001
RGR (%)	141.26 ± 5.11 ^b^	195.58 ± 10.57 ^a^	198.56 ± 8.29 ^a^	210.61 ± 8.03 ^a^	203.01 ± 16.36 ^a^	0.001
SGR (%/day)	1.47 ± 0.03 ^b^	1.80 ± 0.06 ^a^	1.82 ± 0.04 ^a^	1.88 ± 0.04 ^a^	1.84 ± 0.07 ^a^	0.001
FCR	1.76 ± 0.05 ^a^	1.50 ± 0.09 ^b^	1.53 ± 0.11 ^b^	1.45 ± 0.06 ^b^	1.46 ± 0.04 ^b^	0.001

The data presented (mean ± SD) with various superscripts such as a and b are reflected significantly differences (*p* < 0.05) (n = 3). RGR: Relative Growth Rate, SGR: Specific Growth Rate, FCR: Feed Conversion Rate.

**Table 4 animals-12-03387-t004:** Hematological parameters of common carp fed with experimental diets supplemented with different ratios of tomato paste by-product extract (TPE).

	TPE0	TPE0.5	TPE1	TPE2	TEP5	*p*-Value
RBCs × 10^6^ mm^3^	1.13 ± 0.15 ^b^	1.54 ± 0.28 ^a^	1.68 ± 0.19 ^a^	1.60 ± 0.25 ^a^	1.56 ± 0.24 ^a^	0.003
Hct %	30.35 ± 1.77 ^b^	34.95 ± 2.75 ^a^	35.35 ± 2.28 ^a^	35.13 ± 2.58 ^a^	35.25 ± 1.91 ^a^	0.003
Hb g dL^−1^	5.17 ± 0.92 ^b^	6.73 ± 0.54 ^a^	6.89 ± 0.56 ^a^	6.40 ± 0.49 ^a^	6.85 ± 0.74 ^a^	0.001

The data presented (mean ± SD) with various superscripts such as a and b are reflected significantly differences (*p* < 0.05) (n = 6). Hct: hematocrit; Hb: hemoglobin concentration; RBC: red blood cell count.

**Table 5 animals-12-03387-t005:** Blood biochemical parameters of *Cyprinus carpio* fed with experimental diets supplemented with different ratios of tomato paste by-product extract (TPE).

	TPE0	TPE0.5	TPE1	TPE2	TPE5	*p*-Value
Glucose (mg/dL)	114.06 ± 7.28 ^a^	91.69 ± 10.58 ^b^	74.85 ± 5.29 ^c^	66.26 ± 6.54 ^d^	61.83 ± 5.47 ^d^	<0.001
Total protein (g/dL)	7.19 ± 0.95 ^b^	7.16 ± 0.40 ^b^	7.96 ± 0.31 ^a^	7.98 ± 0.27 ^a^	7.86 ± 0.47 ^a^	<0.001
Albumin (g/dL)	0.28 ± 0.06 ^c^	0.51 ± 0.07 ^b^	0.63 ± 0.04 ^a^	0.64 ± 0.03 ^a^	0.61 ± 0.07 ^a^	<0.001
Globulin (g/dL)	6.91 ± 0.92 ^ab^	6.65 ± 0.41 ^b^	7.33 ± 0.31 ^a^	7.34 ± 0.28 ^a^	7.25 ± 0.92 ^ab^	0.008
Triglyceride (mg/dL)	66.63 ± 12.76 ^a^	61.71 ± 9.93 ^a^	39.97 ± 7.88 ^b^	34.54 ± 7.87 ^b^	35.70 ± 9.14 ^b^	<0.001
Cholesterol (mg/dL)	170.80 ± 47.4 ^a^	139.60 ± 27.36 ^b^	85.48 ± 5.90 ^c^	84.07 ± 6.88 ^c^	81.98 ± 6.67 ^c^	<0.001
Aspartate aminotransferase (U/L)	71.09 ± 13.89	51.10 ± 10.03	52.11 ± 8.99	53.44 ±1 1.40	58.77 ± 14.25	0.053
Alanine aminotransferase (U/L)	7.35 ± 1.47	8.01 ± 1.39	7.54 ± 1.73	7.73 ± 1.43	8.41 ± 1.88	0.804
Alkaline phosphatase (U/L)	32.36 ± 9.71	30.73 ± 10.12	29.47 ± 6.71	29.25 ± 8.26	30.79 ± 8.66	0.974
Lactate dehydrogenase (U/L)	308.50 ± 50.00	283.73 ± 16.25	290.70 ± 30.3	250.80 ± 40.10	262.40 ± 29.50	0.064

The data presented (mean ± SD) with various superscripts such as a, b, c and d are reflected significantly differences (*p* < 0.05) (n = 6).

## Data Availability

All data presented this study are available from the corresponding author, upon responsible request.

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
