# Peer review of "Effects of Tomato Paste By-Product Extract on Growth Performance and Blood Parameters in Common Carp (Cyprinus carpio)"

_animals, 2022, doi:10.3390/ani12233387_

Round 1

Reviewer 1 Report

Overall, this study appears to have been conducted appropriately with significant quantities of data collected. More background information and explanation of other studies is required in order to put these results in the proper perspective. I believe a linear regression model to test the effects of graded levels of inclusion would be more appropriate than ANOVA for this study and significant English grammar, word choice, and sentence structure editing needs to take place.

Suggest changing “saturated” to “satiated” in reference to the feeding schedule.

Lines 80-82 are similar to the explanation of the diet treatments in lines 119-122. The description in line 119-122 is written with better English and easier to understand. Suggest removing the description in lines 80-82.

Section 2.5.1 – suggest briefly listing the analyses conducted so readers do not have to go to the results or discussion or look for a reference in order to quickly understand what was measured. Or are these what is listed in lines 137-143? Section 2.5.2.

Statistics – why was a linear regression model not used since this is an examination of a graded level of inclusion? Why was ANOVA chosen?

Discussion – more explanation of the other studies that did not see positive effects of TPE addition need to be described. What levels of TPE inclusion did those other studies utilize? What statistics did they use to show now differences and what other ingredients in the formulations could have been the cause of not seeing added differentiation with the TPE added? Lines 200-210 repeatedly refer to TEP instead of TPE. There are other instances in tables with TEP is used instead of TPE. Suggest providing the “normal” ranges of the blood parameters tested. Stating that TPE helped regulate or raise or lower or improve these parameters is not helpful without knowing their normal ranges and what is good and healthy versus what levels would be cause for concern.

Author Response

S.N.

Comments

Answer

1

Overall, this study appears to have been conducted appropriately with significant quantities of data collected. More background information and explanation of other studies is required in order to put these results in the proper perspective. I believe a linear regression model to test the effects of graded levels of inclusion would be more appropriate than ANOVA for this study and significant English grammar, word choice, and sentence structure editing needs to take place.

First of all, thank you for your valuable comments all of your marked points has been revised in document. In addition to this Figure 1 has been added in MS. We were used ANOVA because of first target of our study was to investigate the positive effects of TPE, which we produced for our study and presented its content in the document, on common carp. We did not look for an appropriate dose in this study. However, your comment opened our horizons and figure 1 was added to the article as allowed by the data we have.

2

Suggest changing “saturated” to “satiated” in reference to the feeding schedule.

The main document has been checked and revision approved.

3

Lines 80-82 are similar to the explanation of the diet treatments in lines 119-122. The description in line 119-122 is written with better English and easier to understand. Suggest removing the description in lines 80-82.

The part of M&M section has been revised by your and the other reviewer comments.

4

Section 2.5.1 – suggest briefly listing the analyses conducted so readers do not have to go to the results or discussion or look for a reference in order to quickly understand what was measured. Or are these what is listed in lines 137-143? Section 2.5.2.

The pointed parts have been revised by your and other reviewer’s comments.

5

Statistics – why was a linear regression model not used since this is an examination of a graded level of inclusion? Why was ANOVA chosen?

Its answered in comment 1 answer.

6

Discussion – more explanation of the other studies that did not see positive effects of TPE addition need to be described. What levels of TPE inclusion did those other studies utilize? What statistics did they use to show now differences and what other ingredients in the formulations could have been the cause of not seeing added differentiation with the TPE added? Lines 200-210 repeatedly refer to TEP instead of TPE. There are other instances in tables with TEP is used instead of TPE. Suggest providing the “normal” ranges of the blood parameters tested. Stating that TPE helped regulate or raise or lower or improve these parameters is not helpful without knowing their normal ranges and what is good and healthy versus what levels would be cause for concern.

Discussion part nearly fully revised and your comments has been reflected in revised version.

Reviewer 2 Report

The manuscript presents results that contribute to advancing knowledge in this area.

It needs minor adjustments that are highlighted in the attached text.

Two Keywords that already appear in the title can be substituted to improve reader reach and experimental design can be detailed in the methodology.

Author Response

S.N.

Comments

Answer

1

The manuscript presents results that contribute to advancing knowledge in this area.

Thank you for your motivational comment.

2

It needs minor adjustments that are highlighted in the attached text.

Marked points have been revised in manuscript file.

3

Two Keywords that already appear in the title can be substituted to improve reader reach and experimental design can be detailed in the methodology.

Underlined keywords have been revised in text. Thank you for your valuable comments.

Reviewer 3 Report

The research article by an international group of 6 researchers submitted to Animals attempts to describe the (positive) effects of tomato paste added into the feed of common carp. The idea behind the research is sound and the possible utilization of waste products from other industries to be used as beneficial ingredient in fish feeds is often worth investigation, especially when there is already prior indication that a certain ingredient could induce positive effects in the target animal. The design of this study seems to be correct, but the writing and general presentation is rather vague and unclear, and due to poor writing and English, the ms is not understandable in several places. It will require thorough editing in terms of English and scientific writing. It is hard for me to believe that all authors have read and accepted the final version before submitting, which is one of the basic rules of ethics in publishing science.

The research has been done in dose-response manner by utilizing four different levels of tomato paste, and the 0-control. In most of the measured variables there is no significant difference between the treatment groups, and this is most surprising in growth and FCR, but also in blood parameters. Only glucose and cholesterol seem to have changed along with the dose. However, there is no discussion why the dose had practically no effect on most of the measured variable but even the slight addition of 0.5% of the paste gave the same response than 5%. The authors should give a credible explanation for this finding. Instead, their conclusion is that the addition of 5% would be the recommended, but after reading the article, for me it is unclear how the authors have ended up with such a conclusion.

I attach a copy of the article with my numerous (over 100) comments/highlights, and can be used as a starting point in improving the ms.

Author Response

S.N.

Comments

Answer

1

The research article by an international group of 6 researchers submitted to Animals attempts to describe the (positive) effects of tomato paste added into the feed of common carp. The idea behind the research is sound and the possible utilization of waste products from other industries to be used as beneficial ingredient in fish feeds is often worth investigation, especially when there is already prior indication that a certain ingredient could induce positive effects in the target animal. The design of this study seems to be correct, but the writing and general presentation is rather vague and unclear, and due to poor writing and English, the ms is not understandable in several places. It will require thorough editing in terms of English and scientific writing. It is hard for me to believe that all authors have read and accepted the final version before submitting, which is one of the basic rules of ethics in publishing science.

First of all, thank you very much for your effort and your belief in our work. Your insightful comment helped us to understand that we need to revise the manuscript deeply. All your comments have been carefully considered and detailed in the following.

2

The research has been done in dose-response manner by utilizing four different levels of tomato paste, and the 0-control. In most of the measured variables there is no significant difference between the treatment groups, and this is most surprising in growth and FCR, but also in blood parameters. Only glucose and cholesterol seem to have changed along with the dose. However, there is no discussion why the dose had practically no effect on most of the measured variable but even the slight addition of 0.5% of the paste gave the same response than 5%. The authors should give a credible explanation for this finding. Instead, their conclusion is that the addition of 5% would be the recommended, but after reading the article, for me it is unclear how the authors have ended up with such a conclusion.

At the end of the first round, you will see a new version of the article in which many statements have been edited and misspellings have been revised to improve clarity.

3

I attach a copy of the article with my numerous (over 100) comments/highlights, and can be used as a starting point in improving the ms.

Your comments which attached by file have been carefully revised in MS. Some of comments need extra information, for that reason selected comments have been answered following list. The MS has more quality with your positive comments and your effort. I am grateful for your improving comments.

Selected comments from attached file

Answer

1

according to table 1 it seems that you have decreased the amount of corn starch accordingly.

All raw materials used in the experimental feeds were analyzed beforehand for the determination of nutrients. Sections supporting this statement were included in the article. Corn starch used in the experimental feeds was used as a binder and its reduction did not show any difference in the protein and energy values of the experimental feeds.

2

was there a certain ratio of FO to TPE? why was it added with FO?

The supplementation constitutes at most 5% of the total ration. For this reason, it was added to the fish oil to be used in feed production in order to provide a homogeneous mixture.

3

you must give details how and where the nutr.content was analyzed

The official nutrient analyses method has been added in that section.

4

Give details for which purpose the diet were made. You must tell the reader what TPE refers to. You must correct all table titles!

All table titles has been revised and TPE has been refers all of them.

5

How did you calculate it? Did you collect uneaten pellets, and if you did, how did you collect them and estimated the uneaten amount?

Glass aquariums were used in the study and feeding was done with high precision. The behavior of the fish was carefully watched during feeding and there was no uneaten feed throughout the feeding experiment.

6

Instead of referring to [26] please give the formulae here.

formulas have been added.

7

how much blood did you withdraw? you let the fish to recover after sampling?

Blood sampling was performed as a terminal application. fish were not included in the system again after blood sampling.

8

How did you analyze all these? What does this reference here mean? Very unclear chapter.

Biochemical measurements were performed using commercial kits. The instructions sent with the kits were followed for the measurements. This section has been revised for clarity.

9

In statistic section, did all data meet these assumptions?

Yes, all data has been showed homogeny and normal distribution.

10

what is n?

Report also what you used as the level of significance.

The significance levels and n numbers have been added as footnotes below the each table.

11

This is not really a result of the experiment but belongs to M&M section.

That part has been transfer under M&M section

12

In M&M you write that 12 fish were sampled per treatment. Why did not you use tank average? These calculations should be explained in M&M the reader to understand what has been done.

Blood was drawing from 12 fish per group in a study with 3 replicates, 4 fish from each tank. sampled blood from 6 fish was used for hematology and the remaining 6 fish blood were used for serum biochemistry parameters. The relevant explanation has been added to the M&M section and n have been added in the footnotes of the each tables.

13

when looking at the table 5 the result does not look quite this straightforward. In some cases there is no difference between 2 treatments. Also, it is totally unclear for the reader why an increase in one value would mean "improvement" while a decrease means "improvement" with some other variables.

The subsection “3.3 Serum biochemical parameters” has been completely revised. More detail has been reflected about Table 5 in the revised text.

14

details of the table 5 are missing

The details have been added at the end of the table 5

15

How did you end up in such a strange conclusion? (in the first sentence of the discussion) All treatment groups were similar. In addition, you have not reported weight gain at all!

Thank you for your contribution with underlined that miswritten sentence. The first part of the discussion has been deeply revised and now the current explanation is more understandable and reflects the study results.

Round 2

Reviewer 1 Report

The authors have adequately addressed the majority of issues raised in the review, would suggest a readthrough for English grammar and sentence structure if possible.

Author Response

We thank the reviewer for the positive comment. Language editing of manuscript has been carried out by native speaker and grammar and spelling errors have been fixed.